# Sad Cases and Success Stories: Representations of Multiple Sclerosis in Direct-to-Consumer Pharmaceutical Advertising

Ella Houston

Centre for Culture & Disability Studies, Liverpool Hope University, Liverpool L16 9JD, UK; houstoe@hope.ac.uk

**Abstract:** This article examines representations of multiple sclerosis in direct-to-consumer pharmaceutical advertisements televised during 2021 in the United States. Drawing on and developing Cultural Disability Studies theory, it highlights how advertising produced by pharmaceutical companies influences mass understandings, as well as personal experiences of, multiple sclerosis. The application of textual analysis to a small-sample of direct-to-consumer advertisements that promote drug therapies for multiple sclerosis (n. 4) uncovers the prevalence of profit-driven, rather than person-driven, medical neoliberal ideologies. On first impressions, the advertisements appear to challenge the metanarrative of multiple sclerosis as a life-limiting tragedy. However, the research findings reveal that multiple sclerosis is framed as the "hidden enemy" of the American dream, supposedly threatening individuals' abilities to live productive and meaningful lives, while the consumption of pharmaceutical "wonder" drugs is treated as an act of self-empowerment.

**Keywords:** multiple sclerosis; direct-to-consumer pharmaceutical advertising; metanarratives; representations; Cultural Disability Studies; popular culture; neoliberalism; medical neoliberalism

## 1. Introduction

Representations of multiple sclerosis (a lifelong neurological condition that impacts the brain and spinal cord) across popular culture invoke the most dramatic stereotypes surrounding disability. Even though it is the 'most common non-traumatic neurological cause of impairment for younger adults' [1] (p. 375), people with multiple sclerosis are underrepresented in the mass media. The few portrayals that exist (usually found in charity advertising campaigns) are almost always designed to grab attention and stir emotions, framing it as a "debilitating disease" striking people in their prime. Disabling stereotypes associated with multiple sclerosis shift from one extreme to the other. Fictional texts often imply that it is synonymous with tragedy, with fictional characters who have multiple sclerosis used as 'symbolic vehicles' to inspire feelings of optimism "despite the odds" amongst non-disabled audiences [2] (p. 1). Often referred to as an "invisible" condition (with symptoms not always noticeable to other people), it is frequently used as a metaphor for 'living in-between ability [and] disability' [3] (p. 179), similar to the 'sword of Damocles' hanging over "normal" life [4] (p. 538).

Advertising campaigns for charities and organisations that aim to support people with multiple sclerosis typically provoke fear as opposed to raising awareness. For example, during the 1980s and 1990s, advertising for the Multiple Sclerosis Societies in the U.K. and U.S. used fear-inducing messages such as, "she can touch her baby's hair but she'll never know what it feels like", also encouraging audiences to be "incurably optimistic" about "searching for a cure for this crippling disease of the central nervous system" [5,6]. Even in more recent years, in 2010, advertisements for the Multiple Sclerosis Society of New Zealand featured shocking images of broken dolls and a hunter (with the head of a deer rather than a human) holding a shotgun, symbolising how "multiple sclerosis turns your brain and body against each other" [7]. Portrayals of multiple sclerosis in advertising undermine personal experiences of 'MS [as] simply a fact of [life]' [8] (n.p.). In contrast to

the metanarrative of multiple sclerosis as a life-limiting tragedy, individuals offer counter-narratives describing it as a 'catalyst' that can 'open up [life] in so many different directions', enabling them to develop deeper levels of patience and appreciation of present moments [9] (p. 1236).

Recognising advertising's ability to influence mass understandings, shaping societal attitudes as well as impacting the self-esteem of people living with the condition, this article examines representations of multiple sclerosis in recent direct-to-consumer pharmaceutical advertising from the United States (U.S.). High-profile, global pharmaceutical companies (who spend billions promoting prescription-only drugs to the general public) are increasingly promoting drug therapies, which have 'changed dramatically' in recent years, as empowering people with multiple sclerosis to '[live] their lives' [10–12] (n.p.). Seemingly positive representations of people with multiple sclerosis enjoying active, fulfilling lives (as a result of pharmaceutical interventions) can challenge harmful misconceptions, including associations with morbidity [1]. However, the research presented in this article reveals that direct-to-consumer pharmaceutical advertisements are heavily influenced by the damaging metanarrative of multiple sclerosis and profit-driven, rather than person-driven, neoliberal ideologies. The research findings suggest that it is framed as "threatening" individuals' capacities to live productive and meaningful lives, while the consumption of pharmaceutical "wonder" drugs is treated as an act of self-empowerment. Drawing on and developing Cultural Disability Studies theory, the current research examines how neoliberalism's tendency to 'privatise, marketise, and depoliticise' experiences of chronic health conditions, impairments and disability are reinforced in direct-to-consumer pharmaceutical advertising [13] (p. 350). Informed by Cultural Disability Studies' recognition that 'the study of culture does indeed enrich our understanding of disability' as disabled people are tied to culturally constructed 'metanarrative[s] by which all is supposedly explained' [14] (p. 343), this article argues that seemingly progressive pharmaceutical advertisements exacerbate myths surrounding multiple sclerosis.

## 2. Literature Review

### 2.1. The "Make-or-Break" Metanarrative of Multiple Sclerosis

The term *metanarrative of disability* encapsulates the 'grand expansion [of] myths, tropes, stereotypes, and other aspects of the cultural imagination' associated with disabled people [15] (p. xvii). Even though attitudes towards disability are often framed as innocent and instinctual, they are deeply embedded in the meanings and values cultures attach to diverse bodies and minds. An example is the so-called 'Western rational medical view' of "mental illness", in comparison to cultures that treat madness as a religious or spiritual "gift" [16] (p. 22). Additionally, the ways in which deafness is appreciated as a positive, linguistic identity in Deaf culture, in contrast to non-disabled people's taken-for-granted assumptions about hearing "loss" [17] (p. 124). As the metanarrative of disability is almost exclusively 'defined by non-disabled people' [15] (p. xvii), it uncovers cultural prejudices surrounding illness, impairments, and human differences, as opposed to reflecting the first-hand experiences of disabled people.

As well as the reductive metanarrative of disability (exemplified in the problematic notion of "the disabled"), there are also metanarratives associated with specific impairments, health conditions, and mental health issues [15]. A sense of tragedy features strongly in the metanarrative of multiple sclerosis, invoked in the myth that it obliterates the happiness and health of whoever it affects. Ignoring how life with multiple sclerosis 'exists on a continuum' [18] (p. 127), with many people having "good and bad days", pharmaceutical and charity advertisements emphasise how it "ruins" the best years of individuals' lives, when they are 'planning families and building careers' [19] (p. 1816). Moreover, the metanarrative of multiple sclerosis is influenced by contextual factors. For instance, in Kuwait it is linked with the Gulf War (1990–1991) as medical research '[traced] the rising prevalence of MS patients to the 1990s', leading many people in this country to believe it is a 'long-term result' of warfare [20] (p. 149). In industrialised parts of the world that are heavily influenced by

neoliberal values of productivity, privatisation, and the power of wealth, the importance of medical "cures" are emphasised, with patients encouraged to "self-manage" long-term impacts of multiple sclerosis, such as muscle weakness and mobility issues [21]. However, individuals often report that they are left with 'unanswered questions' when discussing self-management strategies with physicians [22] (n.p.), increasing the likelihood that they will search for mediated information promoted in direct-to-consumer pharmaceutical appeals. A complex aspect of the metanarrative of multiple sclerosis is the perception of being 'neither properly well nor properly sick' [3] (p. 178). Given that it is an unpredictable condition (that can include periods of symptoms improving, and then getting worse), individuals are often told that they do not '[look] disabled' by other people (p. 181). When they are physically active or do not appear to be unwell, individuals can be made to feel as though they are "malingering" because they do not comply with the 'monolithic', tragedy-steeped metanarrative of multiple sclerosis (p. 184).

Experiences of societal and healthcare barriers are impacted by different aspects of people's identities, including race, ethnic background, social class, gender, and sexuality. For instance, individuals who have multiple sclerosis and are part of the LGBTQ+ community often feel excluded in support groups that are 'filled with "straight, heteronormative [,] binary"' people, feeling as though they are a 'minority within a minority' [23,24] (p. 5). Moreover, even though across the globe women are 'twice as likely as men' to develop multiple sclerosis [25] (p. 1), many experience being 'fobbed off' by medical professionals who believe that their symptoms are 'psychosomatic' [21,26] (pp. 323, 99). Receiving a multiple sclerosis diagnosis can be empowering for women as their symptoms are finally taken seriously [27]. However, many avoid seeking support, fearing they will be perceived as unable to fulfil gendered roles, such as being "classified as an unfit mother due to [having] multiple sclerosis" (p. 101).

Meanwhile, due to widespread 'systemic racism' in healthcare (with people who are part of marginalised ethnic groups being racially stereotyped, while their symptoms are not taken seriously [28]), 'Black, Hispanic and Latinx people' experience notably 'worse clinical outcomes' in comparison to white people with multiple sclerosis [29] (p. 725). Representations of multiple sclerosis in popular culture give the impression that it is an "equaliser", impacting the 'working class to the 'rich and famous' [4] (p. 537). However, for many people living in countries that rely on privatised healthcare, therapies for multiple sclerosis are a 'financial burden' [30] (p. 372). In the U.S., for example, it is a 'highly expensive disease' associated with average medical costs (which can include payments for health insurance and disease-modifying therapies) of '$88,487' (USD) per individual on an annual basis [31] (n.p.).

Despite the numerous societal barriers facing people with multiple sclerosis, popular culture overwhelmingly highlights stories of individuals who are "battling on" even though they are suffering, or those who are "saved" by blockbuster treatments. One of the most popular portrayals of multiple sclerosis, in the American television series *The West Wing* (1999–2006) [32], was heavily criticised by people living with the condition [33]. The series' main protagonist, President Josiah Bartlet (who has relapsing-remitting multiple sclerosis), has been criticised for promoting '"Hollywood MS" (p. 74). As one person who has multiple sclerosis argues, "we've got a man who apparently works from the early morning and he's always walking home in the dark at night unaffected by the day's activities, seemingly, and that's unbelievable" (p. 74). As well as reinforcing the harmful notion that disabled people need to be hyper-productive in order to be validated, *The West Wing's* portrayal of drug therapy (Betaseron) for multiple sclerosis is unrealistic, failing to mention 'how difficult the drug is to obtain' and underplaying side effects 'such as dry skin, flu-like symptoms, and depression' (p. 74). Hollywood stars who have the condition have also contributed to the metanarrative of multiple sclerosis as debilitating and diminishing. For example, actor Christina Applegate used a mobility cane bearing the message "FU [fuck you] MS" at the 29th Screen Actors Guild Awards (2023). Meanwhile, actor Selma Blair is often praised for her awareness raising and advocacy work, while also posing with her mobility cane

at red-carpet events and, more recently, on the front cover of *Vogue* magazine (2023) [34]. However, she also invokes the metanarrative's emphasis on fear, suggesting that you 'feel trapped, a hostage inside your own skin. You are a stranger to you' [35] (p. 243).

Living with multiple sclerosis can be, at times, disabling in and of itself, leading to painful physical issues (such as muscle spasms and nerve damage) and psychological distress. However, as well as acknowledging 'direct [and] immediate impact[s]', it must also be recognised that understandings of health conditions, illnesses, and impairments are 'profoundly social in nature' [36] (pp. 135–137). The symptoms and issues caused by multiple sclerosis are exacerbated by social factors (such as inflexible institutions and stigmatising attitudes). Disabling social behaviours and practices are shaped by cultural representations that reinforce the one-dimensional metanarrative of multiple sclerosis, forgetting that although medical needs and symptoms are 'certainly part of the picture, [so] too are community, relationships, [and] happiness' [37] (p. 109). Indeed, counter-narratives that challenge the myth of multiple sclerosis as a "personal misfortune" highlight how social aspects, such as being pitied and treated differently by other people, often pose the most difficult barriers to overcome [38].

Popular culture frames multiple sclerosis as an interruption to 'happily-ever-after' stories, haunting characters' journeys to success or tales of romance [2,20] (p. 155). Representations in films, for instance, typically follow the same 'visual flow', beginning with people who can 'move freely, [then] need crutches, are confined first to a wheelchair and finally to a bed' [4] (p. 536). The harmful notion is that they have become "imprisoned" in a state of 'residual existence', with their self-worth and value left behind in their past, non-disabled lives (p. 536) [37] (p. 16). Even portrayals that appear to challenge the metanarrative of multiple sclerosis (featuring characters who are optimistic and successful "despite all odds") promote the idea that disability must be conquered by having a determined mindset [4] (p. 537). Many people living with multiple sclerosis adopt a "can-do" attitude, drawing on 'personal reserve[s]' of courage and 'shaping a stable core in a life wrenched by change and loss, change and loss' [8] (p. 76), [21] (p. 327). However, overinflated "success despite multiple sclerosis" narratives fail to recognise any of the ways in which living with the condition provides valuable experiences that can enrich personal identity and knowledge in and of themselves.

Although "success despite MS" messages seem to be affirmative, they 'short-circuit[s] opportunities for more meaningful apprehensions' [39] (p. 2). Many people 'vehemently [criticise]' approaches that are superficially "positive", contributing to the myth that the symptoms and societal barriers they face can be surpassed by miraculous drug therapy or a strong mindset [4] (p. 538). 'Toxic positivity' reinforces rather than challenges the metanarrative of multiple sclerosis, insinuating that it 'negate[s]' all of the good things in a person's life [40] (n.p.). It solidifies the 'damning division' between the so-called "able-bodied" and "the disabled", giving rise to ideas that people with impairments and health conditions fight internal battles in order to have 'lives worth living' [41] (p. 111). Moreover, as well as pressurising individuals to "defy" their health issues, tokenistic representations exacerbate the ignorant attitudes they encounter, such as 'dismissiveness' from health professionals, family members, and employers [42] (n.p.).

### 2.2. Health Is Wealth: Metanarratives, Miracle Drugs, and Medical Neoliberalism in Direct-to-Consumer Pharmaceutical Advertising

Direct-to-consumer pharmaceutical advertisements draw on the metanarrative of multiple sclerosis, emphasising 'discourse[s] of survival' and associating prescription drugs with notions of overcoming symptoms, self-empowerment, and personal autonomy [43] (pp. 12,13). Medical neoliberal myths in pharmaceutical advertising (for example, the idea that prescription drugs are 'magic bullets' that enable hyper-productivity [44] (p. 70)) often target young adults with health conditions or impairments, pushing them to aspire towards neoliberal "ideals" at any cost. Representations of multiple sclerosis in pharmaceutical advertising continue to grow in the U.S., which is one of the 'heartlands' of neoliberalism,

transforming 'patients [into] consumers' [44,45] (pp. 65, 410). Although neoliberal policies, such as repurposing government-funded public services as competitive businesses, have global influence they are most 'heavily applied' in the U.S. [46] (p. 272). As a result of neoliberalism's emphasis on self-reliance, a 'lack of financial support from the welfare state' is one of the most common barriers people with multiple sclerosis encounter [1] (p. 381). Neoliberalism's emphasis on the 'free market' (limiting government interventions in public services that are redesigned as profit-making businesses) means that health care in the U.S. is reframed as a 'commodity rather than a social right' [47] (p. 211). Although the U.S. has a government-funded Medicare program covering various costs for health services and treatments for some people who are disabled and/or aged over sixty-five, private health insurance is more prevalent [48]. Even though the Americans with Disabilities Act (1990) is more than thirty years old, claiming to protect people with impairments from discrimination, disabled Americans have significantly 'less access to healthcare' in comparison to the rest of the U.S. population [49,50] (n.p.).

Pharmaceutical companies promote prescription drugs as 'solution[s] to major public health problems' [51] (p. 35), implying that direct-to-consumer advertisements educate the public on the latest developments in medicine, supposedly improving countless lives. For instance, one of the biggest pharmaceutical corporations in the world, Pfizer (who spent USD 2.8 billion on advertising in 2022), suggests that its advertisements help people to 'make informed choices' about their health [52,53] (p. 38). However, direct-to-consumer pharmaceutical advertising is known for promoting "cutting-edge" medicine at the expense of providing unbiased accounts of health conditions and prescription drugs. Advertisements frequently '[skew] pharmaceutical research' in order to sell newly developed medications that offer minimal 'clinical improvement over existing treatments' [51] (p. 33). When audiences are encouraged to feel a sense of fear, with advertisements relying 'heavily on the metaphor of illness as a thief' [54] (n.p.), they are less likely to deliberate over the risks versus benefits associated with drugs. Direct-to-consumer advertising's promotion of prescription drugs as "solutions" to disabled and chronically ill people's problems, subtly stigmatising those who "fail" to take advantage of "quick-fix" opportunities, warrants critical analysis because they are perceived by many people as trustworthy 'sources of health information' [55] (p. 3). Given that they often refer to clinical trials, some people mistakenly believe that direct-to-consumer pharmaceutical advertisements are reviewed by the government, who only allow medication that is "completely safe" to be promoted [56] (p. 46). Even people who have health insurance enabling them to access expensive medication can feel let down by persuasive appeals that, besides featuring brief statements on major risks, fail to bring attention to the percentage of individuals who experience no improvements or harmful side effects from advertised drugs.

Despite promoting the 'rhetoric of patient activism' and empowerment [51] (p. 37), direct-to-consumer pharmaceutical advertising reinforces the neoliberal narrative of 'strivers and skivers' [57] (p. 292). Glossing over the side effects and overembellishing the benefits linked with prescription drugs, it is as though chronic health conditions and disabling barriers can be so easily overcome [58]. The implication is "golden opportunities" would only be missed by those who are not "thirsty for change". Given that they urge 'patient-consumer[s]' to "empower" themselves by discovering more about the latest prescriptions drugs "without delay" [55] (p. 4), direct-to-consumer advertisements exacerbate the 'emotional blame and shame' that can be internalised by disabled people when they do not meet neoliberal markers of "development" [57] (p. 291). The psychological and emotional impacts brought about by the metanarrative of multiple sclerosis are reinforced, as individuals are pressured to reclaim their "lost lives" at any cost. "Before" images of 'isolation' and suffering are often contrasted with scenes depicting individuals who are "finally" able to enjoy relationships and fulfilling activities, as though disability has been '[wiped] away' [54] (p. 62). As well as encouraging audiences to compare themselves with 'ideal "patient-consumer[s]"' [55] (p. 6), who praise prescription drugs for enabling them to "live life to the fullest", advertisements use upbeat music and persuasive images of 'health, activ-

ity, and strength' [59] (p. 152). Advertisements often feature actors completing rewarding physical and outdoor pursuits, even using images of animals (such as a 'soaring eagle [or] a leaping whale') to convey notions of vitality (p. 152). Although they seem to be positive, these images have guilt-tripping effects, implying that people with health conditions and impairments can easily change their circumstances if they decide to initiate help.

Representations of multiple sclerosis in direct-to-consumer pharmaceutical advertisements reveal how public and self-stigmatisation is shaped by 'neoliberal governance', which supposedly "enables" individuals to control their own successes despite any barriers they encounter [60] (p. 921). Pharmaceutical advertising's rhetoric of empowerment is a double-edged sword—as well as being reminded of their rights to 'make choices about health care', audiences are also persuaded to take advantage of 'whatever products [are available to] treat illness and disease' [44] (p. 68). One of the most common tropes in direct-to-consumer pharmaceutical advertising, which is that 'a healthy appearance and active lifestyle is only a prescription away' [59] (p. 151), reflects the psychological aspects of neoliberalism. As well as shaping social policies and practices, neoliberal ideology '[cultivates] the qualities, dispositions, and feelings needed to survive' in industrialised, profit-driven societies [61] (p. 131). In particular, the neoliberal mindset involves 'radical abstraction of self from context, an entrepreneurial understanding of the self as an ongoing development project [and] an imperative for personal growth and fulfilment' [62] (p. 190). Internalising neoliberal pressures to work harder, faster, and more efficiently can cause individuals to view illness as a 'threat to identity', creating the impression that 'controlling one's illness means controlling one's identity' [59] (p. 137). Alongside being associated with personal fulfilment, pharmaceutical intervention is portrayed as "saving" people from becoming "burdens" on their families and wider society. Indeed, the 'seductive pitch of big Pharma' is that, as well as improving health, medication leads to personal and social 'rewards' such as improved relationships with other people and heightened self-esteem, happiness, and confidence [59,63] (pp. 137, 400).

## 3. Materials and Methods

Rather than developing a large-scale, sweeping review of advertisements, this article provides an in-depth analysis of subtle strategies and persuasive messages used by pharmaceutical advertisers in portrayals of drug therapies for multiple sclerosis. Led by Cultural Disability Studies' emphasis on the power of cultural ideologies and representations in influencing how societies treat disabled people, this research approaches advertisements as documents of socio-cultural attitudes towards disability. As one of the main aims of Cultural Disability Studies is to explore how disability is shaped by cultural contexts, underpinned by the belief that disabling barriers are inevitably shaped by the meanings that cultures attribute to physical impairments, chronic illnesses, and mental health issues, the present research investigates the beliefs, myths, and feelings pharmaceutical advertisements encourage audiences to associate with multiple sclerosis. The advertising industry is renowned for sensationalism at the expense of accuracy [64]. However, portrayals of multiple sclerosis in direct-to-consumer pharmaceutical advertisements invade the personal and public spaces of masses of people on a daily basis, with significant potential to influence individuals' attitudes as they are constantly exposed to advertising via social media, television, newspapers, magazines, and billboards.

The data analysed in this article are gathered from recent direct-to-consumer pharmaceutical advertisements that promote drug therapies for multiple sclerosis. Textual analysis was applied to a small sample (n. 4) of advertisements that were selected due to their explicit focus on multiple sclerosis and because they were broadcast recently, in 2021, on U.S. television. The sample includes *I'm ready for MAVENCLAD*® (Merck KGaA, Roway, NJ, USA, often known as Merck) [65], *Dear MS: Can't Own Us* (advertising Ocrevus, on behalf of Genentech, South San Francisco, CA, USA) [66], *Dramatic Results. Less RMS Drama* (Kesimpta, Novartis AG, Basel, Switzerland) [67], and *I'm Still Me* (Vumerity, Biogen Inc., Cambridge, MA, USA) [68]. Alongside New Zealand, the U.S. is the only country

where it is legal to advertise prescription-only drugs to the general public [69]. While direct-to-consumer pharmaceutical advertising must abide by the U.S. Food and Drug Administration's stipulation that advertisements should not exaggerate potential benefits or underplay risks associated with medication, 'emotive [appeals] linking medicine use with happiness and social approval' are commonplace [70] (p. 1175). Recognising that textual analysis is 'always contextual and thus cultural' [71] (p. 5), this research considers how socio-cultural issues facing people with multiple sclerosis (such as stigmatising attitudes, misconceptions, and pressures to "get better") are reflected in direct-to-consumer pharmaceutical advertising. Instead of producing an 'unassailable representation of the way things "are"', textual analysis explores how texts reveal and respond to socio-cultural ideologies, hierarchies, and practices [72] (p. 220).

Given that television advertising is multimodal and persuasive, data analysis encompassed written and spoken messages, music, colours, images, and scenery, alongside studying characters' expressions and behaviours. Working with a focused sample allowed for detailed notes for each advertisement to be compiled, describing overarching storylines, characters, and the different aspects of scenes. While notes gathered during the first viewing of each advertisement focused on broad themes and plotlines, subsequent viewings generated intricate notes on advertising content (with timestamps to ensure that notes could be double-checked). Rather than organising notes about different elements of each advertisement separately, notes were combined in order to support the identification of overarching themes for individual advertisements in the first instance, followed by the identification of themes across the sample. Features that appeared repeatedly across the data were grouped into initial themes. For example, bright colours, happy facial expressions, and motivational statements were grouped into the initial theme of 'positive and uplifting appeals'. In order to facilitate a critically informed, in-depth analysis of the data, the initial themes were refined into two key themes (examined in the following sections) that capture the main ways that multiple sclerosis is framed across the advertisements. Throughout the data analysis, alongside considering how pharmaceutical advertisements may influence societal attitudes, the ways in which people living with the condition (who 'often seek out mediated information', especially when they are first diagnosed [33] (p. 71)) might be impacted by advertising content is discussed. Analysing the content of self-proclaimed "self-help" texts (which is how direct-to-consumer pharmaceutical advertisements are typically styled, addressed to those who "seek change" and want to "find answers" to their problems) is especially important, considering that the importance of "self-management" is 'enshrined' in medical guidance for people with multiple sclerosis [21] (p. 319).

## 4. Results and Discussion

### 4.1. The Hidden Enemy of the American Dream

One of the most interesting findings of this research is that the sample of direct-to-consumer pharmaceutical advertisements appear to challenge the doom-ridden metanarrative of multiple sclerosis. Rather than depicting people with multiple sclerosis enduring 'residual existence' [37] (p. 16), they are shown leading happy, busy lives (as a result of pharmaceutical interventions). All of the advertisements feature young people with multiple sclerosis enjoying their day-to-day lives, socialising, spending time with their partners and/or children, and engaging with hobbies. In doing so, they raise awareness of multiple sclerosis as a lifelong condition, rather than a "life sentence" [73] (n.p.). Ocrevus' advertisement (*Dear MS: Can't Own Us*), for example, demonstrates how direct-to-consumer pharmaceutical advertising sells "happiness" as well as "fear". Warm and inviting scenes show a young, white woman sharing a blanket with a companion as they huddle by a fire in the forest; a Black man enjoying a meal outdoors with his family; and a white, young woman walking around an urban area (she uses a mobility cane), taking a book from a public lending library and reading it while sitting on a bench. Apart from quick glimpses of the woman's mobility cane, the people who are featured do not present any visible signs of multiple sclerosis (in fact, the woman's mobility cane appears to be the

only "sign" of disability across all of the advertisements). The individuals depicted in Ocrevus' advertisement (described as "real people with MS") model the neoliberal 'ideal "patient-consumer"' [55] (p. 6), suggesting that consuming pharmaceutical drugs enables them to independently steer their own futures (as one person remarks, "I found a way to do things differently with Ocrevus"). The assumption that these people might otherwise be "burdens" on their families and within society more broadly (which is a key element of the metanarrative of multiple sclerosis) casts a shadow over their stories of success. Rather than challenging disabling stereotypes, celebrating individuals who have all of the trappings of the American dream as a result of "miracle drugs" reinforces the metanarrative of multiple sclerosis as a "cruel disease" that threatens to encroach on the best years of people's lives if they do not fight against it.

Multimodal strategies (including appealing images, personalised fonts, bright colours, and uplifting music) enable the advertisements to communicate subtle, engaging, and persuasive messages about prescription drugs. All of the advertisements use light, cheerful music; youthful and confident narrators (who are almost always women); and cute, cartoon-style animations and vibrant colours, emphasising how young people are living life to the fullest now that they have supposedly eradicated the symptoms of multiple sclerosis. The suggestion that it is necessary to be "symptom-free" and "feel better" in order to lead a valuable life exemplifies ableist attitudes that disabled people encounter on a day-to-day basis [54] (p. 63). Instead of paying attention to societal barriers and recognising disability as an aspect of identity rather than an individual "flaw", ableism is invoked when people are encouraged to "cure" or at least downplay their chronic health conditions and impairments in order to 'pass as non-disabled' [74] (p. 301). The advertisements' use of positive appeals, emphasising the "rewards" associated with "curative" prescription drugs, masks ableist ideology that frames disability as the main barrier that needs to be overcome in the quest for a dream lifestyle.

Mavenclad's advertisement (*I'm ready for MAVENCLAD*®) focuses on the benefits associated with the drug rather than the "problem" it targets, with the first scene featuring a young, white woman looking directly towards the camera. As she smiles, the message "I'm ready for" (which is also spoken by the narrator) appears across the screen in bold, capitalised writing, with *ready* highlighted in bright white (contrasted with the orange font used for *I'm* and *for*). The notion that taking Mavenclad indicates she is ready to enjoy the prime of her life, as she is shown wheeling a suitcase through an airport, later on embracing a friend, and taking a 'selfie', communicates the message that disability can only ever be an acceptable part of life if it is suppressed. Moreover, neoliberalism's fixation with self-help and individual motivation is embodied in the suggestion that she is ready to 'take responsibility for [her] health' [55] (p. 6). This advertisement also depicts a woman of colour sitting peacefully as she receives infusion therapy (later on she walks to her job in a school where she teaches children) and a white, young man (wearing a shirt and tie) sitting at his desk, then delivering a presentation to work colleagues. Although these optimistic scenes (accompanied by cheerful background music) challenge assumptions that multiple sclerosis "robs" people's health and happiness, it is implied that success comes easily to those who are "ready for" change. Despite the taken-for-granted notion that disabled people appreciate focusing on the "ability", rather than the "disability", clichéd expressions of encouragement provide the 'perfect breeding ground for forced positivity, decreasing any feelings of genuine compassion' [75] (p. 1). Many people with chronic health conditions and impairments resent reassurances that medical and pharmaceutical interventions mean it is possible for them to enjoy "normal" lives, as the underlying suggestion is accepting disability as an aspect of life and identity is a "meaningless" endeavour. Although the idea that 'defenceless victims' have transformed into 'active people who confront the disease and take their fate into their own hands' seems to be positive [4] (p. 538), it promotes neoliberalism's division between those who "help themselves" and the so-called "helpless". Apart from brief messages about payment support programmes at the end of three advertisements (for Ocrevus, Vumerity, and

Kesimpta), the financial barriers that many Americans with multiple sclerosis face are overlooked, giving the misleading impression that cutting-edge drug therapy is readily accessible for anyone who "seeks change".

Neoliberal society's 'cherry-picking', tokenistic acceptance of the 'able-disabled' [39,76] (pp. 12, 4), whose impairments are treated as supposedly incidental and easily controlled aspects of their lives, is exemplified in Kesimpta's advertisement (*Dramatic Results. Less RMS Drama*). The sole character is a young, white woman who eradicates her symptoms of multiple sclerosis (thanks to Kesimpta) in her quest to lead a happy, busy life. At the beginning of the advertisement she stands in her living room, surrounded by cartoon depictions of common symptoms (for example, a thundercloud represents "active lesions", a crystal ball with a question mark signals "disability progression", and a clock indicates "lost time"), as the narrator asks, "who needs that kind of drama?" She is also shown taking a box of Kesimpta from her fridge and walking through town, occasionally looking at cartoon stickers (such as a bird, a pair of lips, and a cat) on a streetlamp and brick wall. After working on her laptop outside a cafe, she stands on a balcony, taking pictures of a nearby building. During one scene, when she wants to look at herself in a bathroom mirror, she brushes away cartoon depictions of "MS drama" as she wipes away condensation. Although this scene appears to simply provide a peek into a confident, "happy-go-lucky" woman's life, it encourages the problematic assumption that if you make the most of medical and pharmaceutical treatments there are no excuses for not leading a "normal" life. Beneath the glossy veneer, there is the implication that '"feeling better" is a prerequisite' for being a cool, self-confident, and fulfilled young adult [54] (p. 63). Advertisements that promote "wonder drugs", insinuating that getting better is a matter of individual choice, exacerbate the 'emotional blame and shame' that disabled people experience when they are perceived as lagging behind neoliberal markers of development [57] (p. 291). The idea that "wonder drugs" provide quick fixes is a bitter pill to swallow, especially given that many people with multiple sclerosis have to '[fight] for believability' as they are made to feel as though their symptoms are "all in their head" [77] (n.p.).

Instead of explicitly repeating that pharmaceutical interventions enable people with multiple sclerosis to enjoy rewards in their work and social lives, the advertised drugs are framed as 'peripheral to the lifestyle [promised] to consumers' [63] (p. 390). In other words, likeable characters featured in pleasurable and relatable scenes from day-to-day life provide more captivating appeals, which audiences are less likely to think critically about in comparison to jargon-filled messages about prescription drugs. For instance, in Mavenclad's advertisement, there are no images of the drug itself. While uplifting music is used (a technique that takes focus away from obligatory statements about risks and side-effects [63]), there is a lack of sound in the scenes themselves, encouraging audiences to pay closer attention and feel more involved with the advertisement. For example, in scenes depicting a woman teaching school children, although she appears to be talking there is no sound, encouraging the audience to pay closer attention to her body language and movements. Without being stated explicitly, the message is that she is confident and in control of her life as a result of taking Mavenclad. As well as being persuasive promotional strategies, these techniques reflect neoliberalism's 'radical abstraction of self from context' [62] (p. 190), immersing audiences in individualised stories of success. With no scenes featuring the characters' explanations of the ways in which Mavenclad has improved their lives, it is as though the "wonder drugs" speak for themselves. Although the 'distorted' metanarrative of multiple sclerosis is often portrayed explicitly in popular culture, with symptoms 'visualised in detail, often repeatedly' [4,78] (pp. 536, 26), it is promoted in far more subtle ways in pharmaceutical advertisements, as the "hidden enemy" to persuasive narratives of self-empowerment and good fortune.

## 4.2. Selling Self-Empowerment

All of the advertisements imply that they are promoting "miracle drugs" that enable self-empowerment. Not only do they draw on 'discourse[s] of survival' [43] (p. 12), sug-

gesting that pharmaceutical drugs help people to "overcome" the symptoms of multiple sclerosis, they emphasise *thriving* rather than merely surviving. The advertisements reflect society's preoccupation with 'confidence culture', shaped by the neoliberal notion that consumerism, as well as 'self-governance and self-improvement', boosts individuals' chances of being successful [79] (p. 2). As well as communicating this message through non-verbal techniques, including close-up shots of people with multiple sclerosis looking confidently at the camera, alongside scenes where they are featured as the sole or leading characters, the advertisements use 'trigger words' to evoke 'dreams and desires' [80] (p. 149). Emotive words and expressions, such as the suggestion that Kesimpta "may help you put this RMS drama in its place", alongside actors' assertions that "I found a way to do things differently" and "sorry MS, you don't get to control every part of me" (Ocrevus), persuade people with multiple sclerosis to imagine how pharmaceutical drugs could dramatically alter their lives and self-esteem. Emotion-provoking, figurative language also enables direct-to-consumer pharmaceutical advertisements to make claims that 'cannot be easily checked upon', selling dreams and desires to audiences, instead of encouraging them to evaluate the "pros and cons" of interventions (p. 149). The promotion of "wonder drugs" associated with the 'promise of future returns' reflects neoliberalism's 'future oriented cost-benefit calculus', as individuals are persuaded that consuming expensive products, a determined mindset, and hard work is the formula for future success and self-pride [81] (p. 40).

Since audiences are 'drawn to [the] human aspects' of advertising, paying attention to the ways that actors 'communicate feelings, social meanings [and] values' [80] (p. 96), almost all of the scenes in the advertisements focus on actors who appear to be carefree while at home or outdoors, or they are depicted "in action" at work. However, these scenes invoke 'reductive negativity in favour of complexity' [37] (p. 109). Taking a similar approach to charity advertising that claims to "raise awareness" of cancer, despite being 'laced with terror as well as battle language' [82] (p. 169), 'triumph over adversity' is promoted as the only acceptable narrative of multiple sclerosis [83] (p. 5). Instead of being representative of individuals' multi-dimensional, personal experiences (which often involves accepting the condition and its associated symptoms as aspects of life [18]), the advertisements imply that self-fulfilment can only be achieved by "fighting against" multiple sclerosis. In particular, Vumerity's advertisement (*I'm Still Me*) emphasises the idea that pharmaceutical interventions restore individuals' "lost" identities. The advertisement's catchy, memorable jingle connects identity ("I'm still me") with Vumerity ("don't tell me who to be, I'm still me, Vumerity" and "I'm still me, I'm still me, I'm still me, Vumerity"). The association of Vumerity with self-empowerment is reinforced in the advertisement's depiction of a diverse succession of adults, beginning with a young, white woman (who has a tattooed arm) playing with her dog; an older Black woman potting plants with her son; and a young, white man spending time with his husband (wedding bands are visible), as he finishes working on his laptop and then helps to prepare a meal. A small container of Vumerity pills appears in various scenes, such as on the kitchen table as the couple prepare food, while the young woman slips it into her bag before walking her dog. The advertisement reflects the 'universal consumer-citizen', shaped by the neoliberal myth that no matter how different people are, they can all benefit from consumerism and committing to self-empowerment [76] (p. 5). The implication is that despite their different lifestyles and appearances, Vumerity is the key to 'identity control and self-worth' for every single one of the actors [59] (p. 137).

Multiple sclerosis is portrayed as the nemesis of the good, pleasurable things that feature in the advertisements. For instance, Ocrevus' advertisement, *Dear MS: Can't Own Us*, suggests that multiple sclerosis can overwhelm people's identities and lives, as the narrator says: "Dear MS, from day 1 you tried to define me, but I never invited you in. It's my life and this is my journey [ . . . ] Sorry MS, you don't get to control every part of me. MS can't own us". Taking a similar approach to Vumerity, Ocrevus' advertisement focuses on diverse, individual stories. Although the advertisement focuses on three different people (a Black man enjoying a family meal, a white woman on a forest camping trip, and another

white woman strolling around town), they are connected by the same message, "MS can't own us". Neoliberalism's construction of the self as an 'ongoing development project' is emphasised in phrases, such as "you tried to define me", "this is my journey", and "I found a way to do things differently", alongside close-up shots of the characters' determined and self-assured expressions [62] (p. 190). The actors model "go-getting" mindsets that are '[necessary for] survival' in capitalistic societies [61] (p. 131). After putting two fingers up (like a peace sign) to highlight that Ocrevus is only taken twice per year, one actor then holds up a V-sign as she walks away from the camera while the narrator remarks, "sorry MS". However, many counter-narratives of multiple sclerosis describe "make or break" approaches as harmful and misleading, given that acknowledging symptoms and bodily limitations is a key element of acceptance [84].

Even though direct-to-consumer pharmaceutical advertising is typically styled as 'educating' audiences about effective treatments for health conditions, the advertisements mostly focus on building personalised and captivating narratives "proving" the efficacy of pharmaceutical drugs. Developing critiques of direct-to-consumer pharmaceutical advertisements as 'deceptive texts' [85] (p. 180), the research presented in this article finds that they also exacerbate psychological and emotional forms of disablism for people who have multiple sclerosis. Since neoliberal societies value citizens based on their perceived contributions to workforces and economies, the advertisements exploit existing pressures on people with multiple sclerosis to keep on "pushing through, pushing through, pushing through" [84] (p. 238). In the words of one individual, portrayals of "Hollywood MS" "[encourage others to assume that] I'm exaggerating my symptoms for sympathy or am just being lazy" [33] (p. 76). Many disabled people are significantly impacted by the 'psycho-emotional costs of living with ubiquitous cultural 'messages' [about] what is to be valued and what is to be despised' [86] (p. 50). Put differently, the advertisements' narrow conception of what constitutes a "good life" for people living with multiple sclerosis (always being optimistic and ready to make changes happen, while striving towards being active and productive) implies that people who have different experiences (experiencing pain or needing to balance activity and rest) are inherently "lacking".

## 5. Conclusions

This article advances Cultural Disability Studies' understandings of the subtle and persuasive strategies that advertisers use to communicate messages about disability. The research findings reveal that direct-to-consumer pharmaceutical advertisements offer deceptively positive representations of multiple sclerosis. The advertisements analysed in this research appear as though they challenge the tragic metanarrative of multiple sclerosis by focusing on individuals' go-getting attitudes and active, fulfilling lives (supposedly made possible through pharmaceutical interventions). Bearing in mind that advertisements are polysemous, as they are open to diverse interpretations from different people, although they might be praised as "positive" by non-disabled audiences, the idea that having a valuable life and self-identity can only be achieved through drug therapies will be offensive to many people with multiple sclerosis. In a similar way to various portrayals of disability created by advertisers in recent years, the advertisements seem to challenge misconceptions and stigmatising attitudes, suggesting that people with multiple sclerosis can 'compete equally in all facets of life' [87,88] (n.p.). However, they reinforce neoliberalism's tokenistic "acceptance" of the 'able-disabled' who are able to be hyper-productive and work and consume despite having impairments and/or being chronically ill [39] (p. 72). Despite recognising diversity in various ways, featuring Black actors and a gay couple, the notion is that it is desirable for people with multiple sclerosis to disassociate themselves from disability identity. In a way that is typical of seemingly "pro-diversity" representations in neoliberal societies, disability is 'hypermarginalized' as it only considered to be a medical "problem", as something that threatens to make life barely 'worth living' [76] (pp. 4–6). Whereas individuals' personal accounts often discuss accepting multiple sclerosis as part of their identity, suggesting that it teaches them to live in the present and has stopped

them valuing '"doing" over "being"' [9] (pp. 1228), [22], direct-to-consumer pharmaceutical advertisements imply that prescription drugs bring forth a new and more desirable "disability-free" lease of life.

Rather than suggesting that direct-to-consumer pharmaceutical advertising is problematic and unethical in and of itself, this article suggests that pharmaceutical advertisements mislead audiences by implying that prescription drugs help people to reclaim their "lost" identities and lives. As well as over-inflating the benefits associated with drug therapy, this approach also gives rise to harmful myths, stereotypes, and attitudes surrounding illness and disability. The notion that "miracle drugs" give people their lives back promotes the metanarrative of multiple sclerosis as a battle against the self. Meanwhile, counternarratives from people living with multiple sclerosis contend that 'my disability itself is not a sickness. It's part of who I am. And I'm far more likely to thrive if you don't regard me as sick at my very core' [89] (p. 7). Besides undermining personal experiences, direct-to-consumer pharmaceutical advertisements are likely to negatively impact processes of identity development for people with multiple sclerosis. It is accepted that prescription drugs play a crucial role in many people's lives, enabling them to manage symptoms that cause pain and discomfort, healing some medical conditions. However, a key problem challenged by the Disabled People's Movement is the 'assumed authority' of pharmaceutical and medical authorities who 'tell us how we should live' as they presume 'how awful it is to be disabled' [15,90] (pp. xvi, 5). Instead of patronising promises that pharmaceutical interventions lead to "new and improved" lives, people who have impairments and chronic health conditions are more concerned with mass messages about the "problems" they face being fair and realistic. As highlighted in this article, the vast majority of people with multiple sclerosis are more concerned with inaccessible environments, prohibitive and privatised healthcare, and assumptions that their lives are "lacking" than they are with changing who they are.

**Funding:** This research received no external funding.

**Institutional Review Board Statement:** Not applicable.

**Informed Consent Statement:** Not applicable.

**Data Availability Statement:** The data presented in this study are available within this article.

**Acknowledgments:** The author is very grateful to David Bolt for his mentorship and for his feedback on this article.

**Conflicts of Interest:** The author declares no conflict of interest.

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
