# Peer review of "Sad Cases and Success Stories: Representations of Multiple Sclerosis in Direct-to-Consumer Pharmaceutical Advertising"

_societies, doi:10.3390/soc13070158_

Round 1
Reviewer 1 Report
This essay argues that representations in advertising media (particularly those promoted by Big Pharma) depict individuals with Multiple Sclerosis as only needing medication in order to take command of their lives and improve their ability to pass as non-disabled (or at least disabled as possible). The author uses David Bolt's argument about metanarratives that overtake the field of representations in key ways and the argument cultivates a critique that moves against the simplistic optimism of advertising culture. First, the article examines what one might call the semiotics of advertising by paying attention to visual, aural, aesthetic, and artistic framings of individuals with MS. Second, the essay points to the unrealistic organizing idea that prescription medications will alleviate symptoms without other interventions. Third, the arbitrariness of disease etiologies is pointed out as a form of "creaming" takes place in pharmaceutical advertising by only choosing to show middle class people enjoying their lives as a result of seizing the reigns of their disease and taking their medications. Fourth, the author convincingly documents the privatized nature of healthcare in the US where it is permissible to advertise medications as a a result of neoliberal principles of profit-taking, yet there is no address of the difficulty of gaining access such as medications. Finally, the primary thrust of the article is an exposé of ways in which media saturation pervasively promotes individuals as personally responsible for their own health and that illnesses such as MS are ultimately their fault if they don't avail themselves of the opportunities afforded by Big Pharma neoliberalizing schemes. My one reservation that I was hoping to find in the essay (although it does not promise this) is that there is no engagement with narratives of MS created by disabled people. Authors such as Nancy Mairs would really enable this argument to go deeper than the more usual cultural studies framework that employs disability as a barometer of social shortcomings without alternatives of insight and interdependency as a way to complicate these metanarrative framings. It's difficult without this alternative network of critique and alternative ethical mappings of the world to find anything beyond the advertising world's portrayals. If I were to re-write this essay this is the primary change that I would include to go beyond the more typical cultural disability studies argument that it offers (no matter how well it is done and this article manages the terrain it occupies quite adeptly).
Author Response
Thank you very much for taking the time to read my paper and for your helpful comments, which have enabled me to improve my paper. I really appreciate your feedback and suggestion of wider reading. I have responded to your suggestion to engage with counter-narratives of multiple sclerosis in multiple places in my paper, below I have highlighted various examples of additions I have made.
Comment:
My one reservation that I was hoping to find in the essay (although it does not promise this) is that there is no engagement with narratives of MS created by disabled people. Authors such as Nancy Mairs would really enable this argument to go deeper than the more usual cultural studies framework that employs disability as a barometer of social shortcomings without alternatives of insight and interdependency as a way to complicate these metanarrative framings. It's difficult without this alternative network of critique and alternative ethical mappings of the world to find anything beyond the advertising world's portrayals. If I were to re-write this essay this is the primary change that I would include to go beyond the more typical cultural disability studies argument that it offers (no matter how well it is done and this article manages the terrain it occupies quite adeptly).
My response:
P.2: 'Portrayals of multiple sclerosis in advertising undermine personal experiences of 'MS [as] simply a fact of [life]' [8] (n.p.). In contrast to the metanarrative of multiple sclerosis as a life-limiting tragedy, individuals offer counter-narratives describing it as a 'catalyst' that can 'open up [life] in so many different directions', enabling them to develop deeper levels of patience and appreciation of present moments [9] (p. 1236).'
P.3: 'Ignoring how life with multiple sclerosis 'exists on a continuum' [22] (p. 127), with many people having "good and bad days", pharmaceutical and charity advertisements emphasise how it "ruins" the best years of individuals' lives, when they are 'planning families and building careers' [23] (p. 1816).'
P.4: 'Indeed, counter-narratives that challenge the myth of multiple sclerosis as a "personal misfortune" highlight how social aspects, such as being pitied and treated differently by other people, often pose the most difficult barriers to overcome [46].'
P.5: 'Many people living with multiple sclerosis adopt a "can-do" attitude, drawing on 'personal reserve[s]' of courage and 'shaping a stable core in a life wrenched by change and loss, change and loss' [52, 53, 54] (p. 327, 76, n.p.).'
P.12: 'However, many counter-narratives of multiple sclerosis describe "make or break" approaches as harmful and misleading, given that acknowledging symptoms and bodily limitations is a key element of acceptance [137].'
P.13: 'Whereas individuals' personal accounts often discuss accepting multiple sclerosis as part of their identity, suggesting that it teaches them to live in the present and has stopped them valuing '"doing" over "being"' [146, 147, 148] (p. 9, p. 1228), direct-to-consumer pharmaceutical advertisements imply that prescription drugs bring forth a new and more desirable "disability-free" lease of life.'
P.13: 'Meanwhile, counter-narratives from people living with multiple sclerosis contend that 'my disability itself is not a sickness. It’s part of who I am. And I’m far more likely to thrive if you don’t regard me as sick at my very core' ([149] (p. 7). Besides undermining personal experiences, direct-to-consumer pharmaceutical advertisements are likely to negatively impact processes of identity development for people with multiple sclerosis.'
Reviewer 2 Report
My suggestion is to clarify when talking about money that you mean U.S. dollars. It also might be good to clarify in which state the U.S. pharma companies are based and if companies use different names in other countries? Just some things that might help international readers.
Around line 279, can you add another category of patient - one who has insurance that will cover the MS med but the med doesn't work for them or has bad side effects. Drug ads never discuss the % of people for whom the drug doesn't seem to help.
Author Response
Thank you very much for taking the time to read my paper and for sharing helpful comments that have enabled me to improve my work. Please find my responses to your comments:
Comment:
My suggestion is to clarify when talking about money that you mean U.S. dollars. It also might be good to clarify in which state the U.S. pharma companies are based and if companies use different names in other countries? Just some things that might help international readers.
My response:
I have clarified the following: ''$88,487' (dollars)'. I have also checked the name of each pharmaceutical company and ensured that the paper includes the name that they use across the globe (clarifications made within the second paragraph of the Materials and Methods section).
Comment:
Around line 279, can you add another category of patient - one who has insurance that will cover the MS med but the med doesn't work for them or has bad side effects. Drug ads never discuss the % of people for whom the drug doesn't seem to help.
My response:
Thank you for bringing this point to my attention. I have added it to the end of the second paragraph in section 2.2.
Reviewer 3 Report
Outstanding, creative analysis.
Author Response
Thank you very much for taking the time to read my paper and offer feedback. Many thanks, also, for your words of encouragement. I appreciate your help.
Reviewer 4 Report
Thank you very much for the opportunity to review this manuscript. This is an incredibly interesting and well-written paper. I have provided feedback focused on clarification, organization, and strengthening the arguments presented.
Introduction:
MS is not the only disease which is subject to damaging advertising, consider referring briefly to additional populations that have had DTC analyses completed, e.g.,
Violet, T.K. Constructing the Gendered Risk of Illness in Lyrica Ads for Fibromyalgia: Fear of Isolation as a Motivating Narrative for Consumer Demand. J Med Humanit 43, 55–64 (2022).
Cole, Kristen L. "Selling a Cure for Chronicity: A Layered Narrative Analysis of Direct-to-Consumer Humira[R] Advertisements." Rhetoric of Health & Medicine, vol. 5, no. 2, spring 2022, pp. 212+. Gale Health and Wellness
Please provide clarification about which Multiple Sclerosis Society you refer to – with an international readership, there may be multiple countries whose MS society is named the same (e.g., “the Multiple Sclerosis Society of Canada” or “The National Multiple Sclerosis Society of the U.S.” would be sufficient).
Methods:
Paragraph 1: Provide a short description/definition of Cultural Disability Studies to distinguish from Critical Disability Studies/Feminist Disability Studies etc.
Paragraph 1: Though I do not doubt that the advertising industry is renowned for sensationalism, perhaps provide some citations of previous analyses demonstrating this. The last sentence in this paragraph does not clarify what MS DTC advertising is any more invasive than any other form of advertising. It seems that the reason this is particularly damaging for this population is related to the ways it influences the public’s beliefs about this disease more than their beliefs about the products being advertised.
Pg 2 line 95, reference has a page number with no reference.
Perhaps my biggest suggestion is to provide a step-by-step description of your textual analysis. Were the advertisements transcribed word for word with image/music/font/color etc. described separately? What were your inclusion criteria for being selected? Why 4? What quality criteria were used?
When you refer to the “self-help” texts, are you speaking about the colloquialisms of the MS metanarratives? E.g., MS doesn’t have me. Please clarify if you are speaking about texts as in the trite sayings vs. texts as larger pieces of work (like a self-help book).
This may be preference, but I believe the paper would flow more clearly if your methods were moved to after your literature review. You can mention your purpose after your introduction, then move to your literature review, then methods and results. Otherwise, the results seem to come out of nowhere.
Literature review:
Page 4 lines 152, 155, are both of these page numbers associated with reference 32?
Page 4 line 178 – clarify whether this is out of pocket or how insurance plays into this $ amount (again, international readership with different health care systems).
Page 5 line 253 – consider deleting “from” (i.e., change to “… transforming ‘patients [into] consumers…”
Results/Discussion: how were decisions made about the organization of the results? This links back to the comments about the description of your methodology. Please ensure sufficient detail is added to allow the reader to know how you arrived at these themes.
I am curious about the naming of theme 1 – The hidden enemy of the American dream. Given your statement about all the advertisements depicting people living happy busy lives, and the subsequent paragraphs of analysis, it seems like the “helpless” referred to on line 388 are the hidden enemy of the American dream, not necessarily MS (since, the DTC advertisements have the solution to that.) Perhaps clarify the naming of this theme or rename? In general, there were many ideas within theme 1 that weren’t explicitly and overtly connected together in a way that made sense given the naming of the theme. Theme 2 was a great example of a clear theme. Consider revisiting theme 1 to clarify what the true commonality across the examples was.
Line 483-487 – This was a really compelling analysis, I think it might be worth bringing up the tension between embracing other marginalized identities (e.g., race and sexuality are implicated in this advertisement) but rejecting the disability identity. Along with this, on line 550 you refer again to “lost identities and lives”. This reinforces that MS is a thief of identity and contributes to a metanarrative of MS as turning your body against you (the warring battle that you mentioned briefly with cancer). This is really important because this narrative disrupts the process of disability identity development. This ties into what you wrote about psycho-emotional disablism as well.
Round 2
Reviewer 4 Report
The revisions have greatly improved the clarity and impact of the paper. I have no further comments.